# Factor Validity and Generic Reliability of the Developmental Coordination Disorder Questionnaire in the Czech Population

**DOI:** 10.3390/children10060990

**Published:** 2023-06-01

**Authors:** Nikol Vlasakova, Martin Musalek, Ladislav Cepicka

**Affiliations:** 1Faculty of Physical Education and Sport, Charles University, 162 52 Prague, Czech Republic; 2Faculty of Education, University of West Bohemia, 306 14 Pilsen, Czech Republic

**Keywords:** DCD, DCDQ, factor analysis, screening instrument, development disorders, skills motors

## Abstract

The Developmental Coordination Disorder Questionnaire (DCDQ) is widely used as a brief parent questionnaire designed to screen for motor coordination in children, aged 5 to 15 years. There is no validated version of the DCDQ for the Central Europe, which could help for first catch of children with motor difficulties, whose amount has been seriously raised. In addition, the World Health Organization recommends the cross-cultural validation of existing instruments, for Loir costs and time consuming, and the availability of instruments in several languages enables therapists to use validated tools with non-English speaking clients. The aim of this study was to validate the DCDQ in the Czech culture in a population of Czech parents whose children were aged six to ten. Using data from 651 Czech parents of children (six to ten years; 7.8 ± 0.8 years), confirmatory factor analyses (CFA) were used. The goodness-of-fit indices CFI = 0.94, TLI = 0.93, and RMSEA = 0.08 supported the original three-factor model of the DCDQ. In addition, the factor loadings of each question discovered in Czech DCDQ were non-significantly different from the original DCDQ. Furthermore, we also found strong between factor correlation; general coordination and control movement r = 0.87 probably measure the same underlying construct. Even though this is in conformity with original DCDQ structure, we suggest that responses in these two DCDQ factors might have violated the local independency and, therefore, could bias the final score. The generic reliability of the individual factors was acceptable and ranged from McDonald ω 0.83–0.88. Results from this study suggest that cross-validated version of the original DCDQ can be considered as sufficiently valid and reliable clinical screening tool for children who have coordination challenges for Czech children aged six to ten.

## 1. Introduction

Developmental Coordination Disorder (DCD) in children is documented by various terms, including clumsiness, developmental dyspraxia, and specific developmental disorder of motor fiction. During the last years, knowledge about children with developmental coordination disorder (DCD) has greatly increased. Often, the motor problems of these children occur in association with social–emotional problems or learning and attentional difficulties [1,2,3]. Generally, parents notice their child’s motor difficulties from an early age [4,5,6,7]. These concerns are not always recognized by professionals [8], and parents can be incorrectly reassured that their child will outgrow their difficulties [9]. It is not until the child enters the school system that motor problems become more pronounced [7]. With no intervention, the movement difficulties continue to interfere with day-to-day functioning until adolescence in the majority of the children, [10] and they may even have considerable educational, social, and psychiatric consequences [11,12]. Intervention is recommended to assist children with DCD as soon as problems are recognized. However, for most children, the diagnosis comes later, rather than sooner [13]. The use of a reliable and valid screening instrument is essential for the early detection of DCD and subsequent intervention. The administration of a motor test is too time consuming and expensive for population screening. Motor questionnaires that are completed by parents or teachers may be a valuable alternative. In addition, they provide information about the child’s motor skills in daily life (criterion B). If a child is positively identified with a questionnaire, subsequent administration of a standardized motor test is required to verify whether criterion A has been met. Currently, there is no effective instrument for the early detection of DCD in the Czech Republic. The DCD Questionnaire (DCD-Q) was developed in Canada for the identification of children with DCD by parents [14]. This questionnaire seems to have potential as a screening instrument. The questionnaire performs very well in transfers to other countries. In the Italian version of the DCDQ, an internal consistency value (Cronbach alpha) 0.94 was present. The Italian DCDQ achieved moderate to high test–retest reliability for 14/15 items and a good diagnostic performance for identifying children with DCD, sensitivity 88% and specificity 96% [15]. In Spain, confirmatory factor analysis supported the original three-factor structure, and internal consistency was excellent (Cronbach’s α = 0.907) [16]. After examining test–retest consistency in Taiwan, Cronbach’s α for the total score was 0.89, and test–retest reliability was 0.94. Exploratory and confirmatory factor analyses showed that this version was compatible with the original and two modifications of the DCDQ [17]. The validation of the DCDQ has had very good results in other countries, such as Germany [18], the Netherlands [19], Brazil [20], Japan [21], and a study by Rivard [22]. However, it is also necessary to mention that, in certain studies, some items from DCDQ were removed with the aim to achieve acceptable fit model values or reliability [15,19,20]. In case of validation in the Netherlands, [19] found, as the best four factors, a model that removes item 11 (“Your child is interested in and likes participating in sports or active games requiring good motor skills”).

Children with DCD often have problems with writing and other academic outcomes; these children [23], therefore, possess a diagnosis of these disorders, which are often not addressed until younger school age. This was the reason we chose to conduct the study at ages six to ten. The choice of a median age norm (six to ten years) was also motivated by the findings of the largest study in 2014 by Rivard [22], showing that older children have significantly higher mean scores than younger children, which is not surprising in the population. The results of this study confirm that this convention is still appropriate, especially for younger children. This was the reason we chose to conduct the study at age six to ten.

In the Czech Republic, the diagnosis of motor disorders, such as DCD, is often not made until the start of primary school. These disorders are often identified only in the school environment and are, therefore, diagnosed only at a younger school age, when their distinctiveness in the collective is highlighted. This fact was also the reason we chose to conduct the study at age six to ten. At this age, it is also possible to start intervention with the help of teachers, and the Czech version of DCDQ can greatly help in diagnosing DCD in the Czech population. So, the aim of this study is to validate DCDQ in Czech culture.

## 2. Materials and Methods

### 2.1. Measures

The Developmental Coordination Disorder Questionnaire (DCDQ) is a 15-item parent questionnaire designed to screen for coordination disorders in children aged 5–15 years, including playing ball (throwing, catching, hitting), and writing (fast, legibly, with proper effort). Parents were asked to provide their responses on a five-point Likert scale when comparing the motor performance between their child and peers. Each item is scored from 1 to 5 points, giving a total score of from 15 to 75 points, and a high score suggests no DCD. The total score indicates whether the child is in the group of children with “indicated, or suspected DCD”, or “probably not DCD”, according to three age groups (5–7 years and 11 months, 8–9 years and 11 months, and 10–15 years) [22].

### 2.2. Translation and Cross-Cultural Adaptation

The DCDQ was translated according to guidelines developed by Beaton and Bombardier [24,25] for cross-cultural adaptation of assessment instruments. The questionnaire was translated twice into Czech, and cultural words and idioms were adjusted to ensure greater comparability across international populations [26]. First, the DCDQ was translated from English into Czech by two Czech mother tongue independent professionals who adapted items to context and culture. The report was written. For example, some items were changed to fit in the metric system (e.g., Item 2—‘Catches a small ball thrown from a distance of 6 to 8 feet’—was changed to ‘Catches a small ball thrown from a distance of 2 m’). The phrase ‘bull in a China shop’ was changed to a more typical Czech idiomatic sentence. Items referring to sports not common in the Czech Republic (e.g., badminton) were changed to more popular Czech ball-based activities. An expert committee, consisting of two translators and an observer, then reviewed both versions, resolving discrepancies and producing a common translation. The DCDQ was then back translated by two English mother tongue professionals blinded to the original version. Some other terms have been slightly modified to capture the given meaning and linguistic correctness, as the literal translation did not meet the intended meaning in the target language. The pre-definitive version was tested with respondents who are fully competent in both languages. A pilot study correlated the skins of the translated version of the questionnaire with the original questionnaire. Two respondents had significantly different answers for the last two questions. These questions are of the reverse type and, therefore, may be slightly confusing. Unfortunately, due to the scale and scoring given, it was not possible to convert them into a positive form to make them easier to understand. Therefore, the negative wording has been bolded to visually alert the respondent to some change in the questioning. The two reverse questions are placed last in the questionnaire, which, in our opinion, is rather unfortunate given the fact that, by the end of the questionnaire, the respondent is already losing focus. Therefore, we also recommend that the respondent be made aware of this fact in advance of completing the questionnaire.

### 2.3. Participants

A sample of typically developing children and their parents was obtained in collaboration with primary schools in the Czech Republic. A self-reported bulk data collection was carried out, resulting in a representative sample where the population studied was children aged 6–10 years. Primary schools were selected on the basis of stratified sampling, and then random sampling was carried out in the schools. According to the power analysis, a Monte Carlo simulation of the data was performed, which showed that, in the case of a respected test power of 0.9 and a number of three indicators, it was necessary to have at least 430 respondents (Muthén and Muthén 2010). The condition for completing the questionnaire was the age of the child being 6–10 years and that these children had to attend a regular primary school. An amount of 850 parents participated in the survey. Informed consent was obtained from all subjects involved in the study. A major milestone in a child’s life is starting school, so a sample of children (younger school age) aged 6 years, 0 months to 9 years, 11 months (n = 651, mean age = 7.8, SD = 0.8) was included in the study. This is a secondary analysis of data obtained from public primary schools in the Czech Republic (n = 17 schools). The questionnaire could be completed by the child’s father or mother. Informed consent was obtained from all subjects involved in the study. The questionnaire was completed by 651 parents, of whom 485 were mothers, 104 were fathers, 23 were teachers, and 39 did not indicate their gender. In future studies, it would be beneficial to ensure complete data collection from all groups to allow for a more comprehensive comparison between mothers, fathers, and teachers. Ethical approval was obtained from school boards, and consent was obtained from all participating students and parents before data collection began. Questionnaires that were not completely filled in or were missing certain information were discarded. After data cleaning, we obtained 651 children, who constitute the current study sample. The ratio of boys to girls was roughly balanced (1:1), (mean age = 7.8, SD = 0.85) and the sample was representative of the socio-demographic characteristics of the population. A major milestone in a child’s life is starting school, so a sample of children (younger school age) aged 6 years 0 months to 9 years 11 months (n = 651, mean age = 7.8, SD = 0.8) was included in the study. Thus, the condition for completing the questionnaire was that the children were aged 6–10 years, and these children had to attend a mainstream primary school. After cleaning the data, we obtained 651 probands, meeting the conditions according to the confidence interval approach [27].

### 2.4. Procedure

After informed consent was obtained from the children’s parents, parents of children in the population-based sample were asked to fill in the DCD-Q at home. To prevent bias, the tester had no prior knowledge of the children’s scores on the DCD-Q.

### 2.5. Statistical Analysis

For data analysis, we used the standard scores of the DCDQ test items. We verified the factorial validity of the MABC-2 with confirmatory factor analysis (CFA). Since the data in DCDQ have categorical ordered character, we applied the weighted least square mean and variance adjusted estimator [28]. The quality of the model was assessed with the following fit indices: (1) model discrepancy: Chi-square (S-BX2), model significance *p* > 0.05; (2) incremental fit indices: Comparative Fit Index (CFI) ˃ 0.90, Tucker-Lewis Index (TLI) TLI > 0.90, (3) approximating error: Root Mean Square Error of Approximation (RMSEA) < 0.08 [28,29,30]. To reveal possible causes of the model’s low fit indices, we checked differences in factor loadings of test items and correlations among factors among each age category.

Except for the fit indices, the differences between the observed and predicted covariances in residual matrices were investigated. Since the multivariate normality of items was rejected, we analyzed values of the standardized residual [31,32], where values higher than 0.100 are considered significant [33]. Generic reliability of each factor was approximated by McDonald ω, with acceptable value > 0.80 [34].

## 3. Results

In Table 1, there are frequencies of responses in each item. Items: 4. “Jump over”, 5. “Run & stop”, 6. “Plan activity”, 11. “Like sports”, and 14. “Elephant in a shop” had high frequency of answer category five (extremely similar to your child), which means that the child has no motor clumsiness or plans movement activity adequately. In other items, we found different distribution of responses due to significantly higher frequency in response category four (a bit similar to your child). In all questions, the frequency of responses pointing on motor difficulties, categories one (not at all similar to your child), and two (a bit similar to your child) were in the range 0.1% to 2%.

CFA applied on the Czech translated version of DCDQ showed acceptable fit of the original three factor model although the received values are on lower end (Table 2). All fifteen items in Czech DCDQ version had very similar, non-significantly different, factor loadings compared to valued achieved form original version. The factor loadings for each indicator ranged from 0.62–0.85 for the original questionnaire and 0.61–0.80 for the Czech version (see Table 3). When we investigated the residual correlation matrix, we discovered that items 11. “Like sports” and 12. “Learning new skills” displayed significant unexplained variance > 0.100 with the following items: 5. “Run & stop”, 8. “Writing legibly”, 9. “Pencil pressure”, 10. “Cutting”, 15. “No fatigue”, and even with each other.

In data analysis, we also checked Mod-indices results. We found that item number 6. “Plan activity” should be multi-factorially related to both GC and FM constructs, and items number 11. “Like sports”, 13. “Quick/competent”, and 15. “No fatigue” should be multi-factorial related to both by FM and CM. Furthermore, we also discovered significant indirectly and directly correlated errors in range r = 0.20–0.60 between items: 1. “Throw ball” and 2. “Catch ball”, 3. “Hit ball”, 5. “Run & stop”; 2. “Catch ball” and 3. “Hit ball”, 4. “Jump over”, 5. “Run & stop”; 3. “Hit ball” and 6. “Plan activity”; 6. “Plan activity” and 7. “Writing speed”, 8. “Writing legibly”; 7. “Writing speed” and 9. “Pencil pressure”; 11. “Like sports” and 5. “Run & stop“, 7. “Writing speed”, 9. “Pencil pressure”, 10. “Cutting”, 12. “Learns new” (Table 4.) These findings showed that, in responses to aforementioned items, the local independency requirement is violated. It means that not all items from DCDQ are conditionally independent and that its relations are explained by something more than estimating of the same trait–factor. Furthermore, violation of the local independency between different items would explain strong between-factor correlations GC and CM, as well as FM (Table 3). Mainly the correlation between CM and GC r = 0.87 shows that both factors measure very similar latent trait. Although we discovered certain discrepancies in the Czech DCDQ three factor structure, we did not modify, nor re-build, the original three factor structure of the DCDQ because the aim of this study was to validate the original version of DCDQ to the Czech population.

In the last step, we estimated the generic reliability of each construct from DCDQ (Table 5) and factor correlations (Table 6). From Table 4, it is evident that all factors estimated the level of motor difficulties with acceptable construct reliability.

## 4. Discussion

DCD is a neuro-developmental disorder that affects motor coordination and affects both children and adults. This study aim was to validate the DCDQ in the Czech population. The version of the DCDQ 07 [24] questionnaire was used for our study; this version, as well as the revised version [14] of the DCDQ, also support a three-factor structure.

Before using CFA, we performed an exploratory data analysis where we found that there were differences between the distribution of responses in several items. “Jump over”, “Run & stop”, “Plan activity”, “Likes sports”, and “Elephant in a shop” showed significantly lower frequency of responses in category four—“a bit like your child”. It can mean that parents were not able to image properly what “a bit like your child” means in certain situation, or that scale for discriminating is not enough sensitive to catch this category of response in relation to the character of question.

CFA showed that three-factor structure, including “control during movement”, “fine motor and handwriting”, and “general coordination” factors has acceptable fit in the Czech environment: RMSEA = 0.08, CFI = 0.94, TLI = 0.93. Furthermore, all DCDQ items had similar, non-significantly different factor loadings, 0.61–0.80, in comparison to the original DCDQ. The final questionnaire, therefore, closely resembled the original [14] (Wilson et al. 2009). Furthermore, the psychometrically meaningful three-factor structure of DCDQ has also been supported by most previous cross-cultural studies [15,17,35,36]. However, when we went through the previous DCDQ cross-cultural studies, we discovered that, in some of studies, the better fit of the model was achieved my removing some items, modifying the place of items in the structure, or making changes in the structure [15,19,20]. In the structural equation modeling, we can obtain almost perfect model fit if we respect the modification indices recommendations, which is, however, based only on mathematical calculations. Therefore, for changes in structural models, strong behavioral support must be present [37,38]. The aim of our study was to validate original DCDQ in a Czech environment, which contains fifteen items in solid three-factor structure. Therefore, we did not adjust the number of items nor correct the structure to achieve the best model fit, such as changing the number of factors to achieve the best model fit as [19,39] did. We assume that respecting the original structure of DCDQ with all fifteen items was one significant cause for the model fit values in our study being on the lower bound of acceptability. Concerning the complete validation process, there is also the question if all previous cross-cultural studies conducted precise translation procedures because, in the majority of them, authors did not describe in detail the translation process. Therefore, in some of studies [20,39], which remove the items or changed the structure of DCDQ, it seemed that semantic issues played a significant role in poor discriminatory power of certain items caused, for instance, by its double negative form [18,40]. Likewise, in our validation study, we had carried out translation of DCDQ to the Czech language, including language stylistic corrections to keep the semantic content of items in separate research [41] before we conducted the present CFA study.

Based on model fit values, we focused on residual correlations greater than 0.100, pointed on a large unexplained portion of a relationship between the empirical and predicted correlation. We discovered two problematic items—11. “Like sports” and 12. “Learning new”. We assume that both these items reflect motivation aspects, rather than direct impact of possible motor difficulties rising from motor performance. To learn new things or to prefer something is driven by personal motivation, which means the interaction of the individual and the situation to move towards the goal [42]. Certain support for this explanation provided the studies pointed that associations between motivation for physical activities and performance in motor competencies seems to be non-significant [43]. Furthermore, positive or negative motivation in sport participation is influence by many of factors between which the most significant is motivational climate. The most significant elements of motivational climate positively influencing sport of physical leisure time participation involve the social unit family, school teachers (particularly PE teachers), and using strategies in PE classes, which may consist of tasks or ego-orientation and mates [44,45,46]. The degree of motivation was found to be associated with perceived motor competencies or with the amount of physical activity [47]. Both items 11. “Like sports” and 12. “Learning new” displayed the greatest unexplained variance, with items from the FM construct assessing involvement in the movement smoothness and accuracy in activities highly demanding fine motor skills (pencil pressure, writing legibly, cutting). Even though the learning process and acquisition of fine motor skills, such as handwriting or cutting, potentially involve the same neural regions—the cerebellum and the left dorsal premotor cortex [48]—the motivation to learn something or to prefer something is built up on more complex central nervous system interactions combining biological, social, emotional, and cognitive aspects [49]. Furthermore, we also investigated the between-item error correlations, which would point to possible items’ local independency violation. This local independency violation was found between the items in CM, which relates to object control with the ball: throwing, catching, or hitting. In addition, another cluster of significant error correlations was found in the items composing the FM factor fine motor activities. The strongest error correlations were between “Like sports” and “Writing legibly” or “Like sports” and “Learn new”. These results show that responses in certain DCDQ can significantly influence the response in another DCDQ. The implication of such conditional influence causes overestimation of the reliability of assessed traits [50], such as the risk of motor difficulties.

Large residual and error correlations between some of the DCDQ items might also be a cause for strong between-factor correlation, which we found. According to the guideline from [51], between-factor correlation > 0.80 shows that two factors measure the same underlying construct or domain. In our study, we found correlation between CM and GC factors r = 0.87. It means that these two factors share a significant portion of variance. In CM constructs, all items asked for movement skill performance—object control with the ball or a sort of planning or self-regulation. In GC, the items ask for the states: “Like sports”, “Learn new”, “behave as Elephant in shop”, or if the person is “shortly Fatigue”. In our opinion, problems with performance in object control or self-regulation in movement are mirroring the state or process–motivation to learn something, or it seems to be clumsy or prefers sports where the proficiency from object control is usually necessary. Based on the results, we suggest that the GC factor might be superior to the CM factor or that the CM factor could be nested into the GC factor.

Reliability: For the approximation of reliability, we calculated generic, construct reliability. The generic reliability calculated as McDonald ω was GC = 0.84, CM = 0.88, and FM = 0.83, respectively, which mean, according to recommendations from structural equation modeling [34,37,51], “good” reliability is attained. Previous studies [15,16,17] found slightly higher values of DCDQ reliability. However, we must note that, in these studies, the specific reliability as internal consistency or test re-test reliability showed that stability of items across time were used. The specific reliability is connected with the error of each manifest variable instead of construct reliability, which points to the error with which the trait is estimated. Since, in the calculation of generic reliability, we have to consider the uniqueness of each used manifest variable related to construct, the generic reliability value will be lower in comparison to the specific reliability calculated for one manifest variable on the basis of items correlations [52]. To our best knowledge, this is the first study which approximated construct reliability of DCDQ and, at least in the Czech environment, the construct (generic) reliability was shown to be satisfactory for assessing the motor difficulties in children aged six to ten years. The findings show that the DCDQ is a reliable and valid tool for assessing motor coordination problems and for identifying children with probable DCD in the Czech context. Healthcare professionals working in pediatric primary care with children, such as occupational caretakers and physiotherapists or educational and psychological counseling staff, can benefit from these findings and use the DCDQ to operationalise the B diagnostic criteria for DCD. This study only addressed the younger age group offered by the original questionnaire (six to ten years), so we recommend that future studies address the 11–15 year age group.

### Limitations

We have no comparison between DCDQ and motor performance in children. The study only looked at children aged six to ten years, a critical age when motor development disorders are often identified due to school entry. However, the questionnaire was able to identify children aged 5–15 years. It may be a topic for further study. While this study provides valuable insights, it is limited by the lack of information on the respondents who completed the questionnaire. In particular, it would be beneficial to distinguish between responses from mothers, fathers, and teachers to enable more detailed comparisons where relevant. Additionally, the findings of sensitivity and specificity would be worth exploring in the future. On the other hand, the results were received from a good representative sample, and the DCDQ was adequately translated into the Czech language with all language and semantic understanding specifics for Slavic languages.

## 5. Conclusions

DCDQ’07-CZ showed satisfactory validity and reliability, supporting the original three-factor structure. Nevertheless, we found significant error correlations between items in CM and FM constructs. It means that response on one item in these constructs influence the response in the next item, which can imply overestimation or underestimation of the final DCDQ score. Furthermore, the items “Like sports” and “Learn new”, which express motivation, were shown to have large unexplained variance, with items pointing to assumed motor performance in certain situations. These findings would also contribute to explanation for strong between-construct correlation of GC and CM and the suggestion that GC seems to be superior for the other DCDQ constructs of FM and CM. Regardless of certain structural lacks, the DCDQ is a promising, simple, and cost-effective instrument that is clinically useful for parents and teachers in children aged six to ten years in the Czech Republic for the identification of motor difficulties and determining the strategies for children’s development.

## Figures and Tables

**Table 1 children-10-00990-t001:** Percentage and frequency of responses of individual items.

		Category				
		1	2	3	4	5
1. Throw ball	Percentage (%)	0	8	7	46	39
	Frequency of responses	2	53	42	298	255
2. Catch ball	Percentage (%)	0	8	11	44	37
	Frequency of responses	2	52	72	285	239
3. Hit ball	Percentage (%)	2	19	22	34	24
	Frequency of responses	10	120	144	220	156
4. Jump over	Percentage (%)	1	3	3	24	69
	Frequency of responses	4	19	22	156	449
5. Run & stop	Percentage (%)	1	5	5	19	71
	Frequency of responses	3	30	31	126	460
6. Plan activity	Percentage (%)	1	1	2	19	77
	Frequency of responses	6	7	15	123	499
7. Writing speed	Percentage (%)	1	5	20	28	47
	Frequency of responses	6	32	127	180	305
8. Writing legibly	Percentage (%)	1	8	4	38	49
	Frequency of responses	7	49	26	247	321
9. Pencil pressure	Percentage (%)	1	12	5	43	39
	Frequency of responses	7	77	32	281	253
10. Cutting	Percentage (%)	1	8	4	43	45
	Frequency of responses	6	53	26	276	289
11. Likes sports	Percentage (%)	1	4	5	21	70
	Frequency of responses	5	24	33	133	455
12. Learns new	Percentage (%)	2	8	10	28	52
	Frequency of responses	11	54	65	180	340
13. Quick/competent	Percentage (%)	1	11	5	39	45
	Frequency of responses	6	73	29	251	291
14. Elephant in a shop	Percentage (%)	2	3	3	21	71
	Frequency of responses	15	21	16	138	460
15. No fatigue	Percentage (%)	2	11	11	35	42
	Frequency of responses	15	70	68	227	270

**Table 2 children-10-00990-t002:** Factor analysis of Developmental Coordination Disorder Questionnaire for population-based group aged six to ten years (n = 651).

	χ2	df	*p*	CFI	TLI	RMSEA
Original 3—factor model DCDQ-CZ	458.79	87	<0.01	0.94	0.93	0.08

**Table 3 children-10-00990-t003:** Factor analysis.

DCDQ Items	DCDQ–CA Original 3Factor	DCDQ–CZ Original 3Factor
Control Movement (CM)		
1. Throw ball	0.85	0.76
2. Catch ball	0.85	0.72
3. Hit ball	0.81	0.66
4. Jump over	0.81	0.80
5. Run & stop	0.73	0.79
6. Plan activity	0.62	0.70
Fine Motor (FM)		
7. Writing speed	0.85	0.77
8. Writing legibly	0.83	0.80
9. Pencil pressure	0.77	0.70
10. Cutting	0.75	0.70
General Coordination (GC)		
11. Likes sports	0.78	0.76
12. Learns new	0.77	0.73
13. Quick/competent	0.75	0.61
14. Elephant in a shop	0.77	0.78
15. No fatigue	0.73	0.67

**Table 4 children-10-00990-t004:** Error correlations between all items from DCDQ.

	Throw Ball	Catch Ball	Hit Ball	Jump Over	Run & Stop	Plan Activity	Writing Speed	Writing Legibly	Pencil Pressure	Cutting	Likes Sports	Learns New
Throw ball	**1**		-------	-------	-------	-------	-------	-------	-------	-------	-------	-------
Catch ball	**0.30**	**1**	-------	-------	-------	-------	-------	-------	-------	-------	-------	-------
Hit ball	**0.20**	**0.33**	1		-------	-------	-------	-------	-------	-------	-------	-------
Jump over	NS	**−0.24**	NS	1	-------	-------	-------	-------	-------	-------	-------	-------
Run & stop	**−0.29**	**−0.26**	NS	−0.31	1	-------	-------	-------	-------	-------	-------	-------
Plan activity	NS	NS	**0.31**	NS	NS	1	-------	-------	-------	-------	-------	-------
Writing speed	NS	NS	NS	NS	NS	**0.28**	1	-------	-------	-------	-------	-------
Writing legibly	NS	**−0.24**	NS	NS	NS	**0.28**	**0.28**	1	-------	-------	-------	-------
Pencil pressure	NS	NS	NS	NS	NS	NS	**−0.43**	-------	1	-------	-------	-------
Cutting	NS	NS	NS	NS	NS	NS	NS	NS	NS	1	NS	NS
Likes sports	NS	NS	NS	NS	**0.36**	NS	**−0.24**	**−0.60**	**−0.37**	NS	1	NS
Learns new	NS	NS	NS	**0.24**	NS	NS	NS	**−0.38**	NS	**−0.27**	**0.40**	1

Correlations written in bold are significant *p* < 0.05.

**Table 5 children-10-00990-t005:** Generic reliability of factors.

Factors	McDonald ω
Control during movement	0.88
Fine motor	0.83
General coordination	0.84

**Table 6 children-10-00990-t006:** Factor correlations.

Construct	Fine Motor	General Coordination	Control during Movement
Fine motor	1	0.72	0.59
General coordination	0.72	1	0.87
Control during movement	0.59	0.87	1

## Data Availability

DOI 10.5281/zenodo.7990843.

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
