# Peer review of "Factor Validity and Generic Reliability of the Developmental Coordination Disorder Questionnaire in the Czech Population"

_children, 2023, doi:10.3390/children10060990_

Round 1

Reviewer 1 Report

Observations:   1. I suggest changing the keywords to words that are not in the title. In this way, you can increase the chances that other researchers will find the study in question.

Keywords: Screening Instrument; Development Disorders; Skills Motors.

  2.  I suggest indicating the composition the sociodemographic characteristics of the participants, and the educational level of the parents. 3. The objective of the study appears written in four different ways: 1) pg 1, lines 17 and 18; 2) pg 6, lines 213 and 214; 3) pg 7, lines 238 and 239; 4) pg 7, lines 262 and 263.

I suggest that the goal be written in a single form. I indicate that the proper form of the objective is:

A) To adapt the DCDQ in Czeck culture or Czeck context or Czeck environment;

OR

B) Validate the DCDQ in the Czeck culture or Czeck context or Czeck environment.

These forms are simple, direct, and clear.

  4. I suggest changing de punctuation signal:

NV: MM, and LC

NV, MM, and LC.

Author Response

Point 1: I suggest changing the keywords to words that are not in the title. In this way, you can increase the chances that other researchers will find the study in question.

Keywords: Screening Instrument; Development Disorders; Skills Motors.

Respond 1: Text added

  Point 2:  I suggest indicating the composition the sociodemographic characteristics of the participants, and the educational level of the parents.

Respond 2: Unfortunately, this data was not collected.

Point 3: The objective of the study appears written in four different ways: 1) pg 1, lines 17 and 18; 2) pg 6, lines 213 and 214; 3) pg 7, lines 238 and 239; 4) pg 7, lines 262 and 263.
I suggest that the goal be written in a single form. I indicate that the proper form of the objective is:

A) To adapt the DCDQ in Czeck culture or Czeck context or Czeck environment;

OR

B) Validate the DCDQ in the Czeck culture or Czeck context or Czeck environment.

These forms are simple, direct, and clear.

Respond 3: Corrected and unified

Point 4: I suggest changing de punctuation signal:
NV: MM, and LC

NV, MM, and LC.

Respond 4: Done

Reviewer 2 Report

Thank you for the opportunity to review this validity and reliability study for the DCD questionnaire.

Some considerations for authors:

- Include the number of suitable keywords and that are mesh terms.

- It has not been justified why it is necessary to carry out this study in the introduction.

- Table 4: "non sig" can be shortened to "NS" and the table is more summarized.

- This current reference can help you to explain well how the DCD works: Pinero-Pinto E, Romero-Galisteo RP, Sánchez-González MC, Escobio-Prieto I, Luque-Moreno C, Palomo-Carrión R. Motor Skills and Visual Deficits in Developmental Coordination Disorder: A Narrative Review. J Clin Med. 2022 Dec 15;11(24):7447. doi: 10.3390/jcm11247447.

- It would have been better to compare the results with another similar questionnaire to verify the results, perhaps it could be included as another limitation, comparing with another scale or tool.

- There are some quite old references, review those that are dispensable from more than 10 years ago.

Author Response

Point 1: Include the number of suitable keywords and that are mesh terms.

Respond 1: Added

Point 2:  It has not been justified why it is necessary to carry out this study in the introduction.

Respond 2: 

Added: At this age it is also possible to start intervention with the help of teachers and the Czech version of DCDQ can greatly help in diagnosing od DCD in the Czech population.

Point 3: Table 4: "non sig" can be shortened to "NS" and the table is more summarized.

Respond 3: done

Point 4: This current reference can help you to explain well how the DCD works: Pinero-Pinto E, Romero-Galisteo RP, Sánchez-González MC, Escobio-Prieto I, Luque-Moreno C, Palomo-Carrión R. Motor Skills and Visual Deficits in Developmental Coordination Disorder: A Narrative Review. J Clin Med. 2022 Dec 15;11(24):7447. doi: 10.3390/jcm11247447.

- It would have been better to compare the results with another similar questionnaire to verify the results, perhaps it could be included as another limitation, comparing with another scale or tool.

- There are some quite old references, review those that are dispensable from more than 10 years ago.

Respond 4: 

The study was done several years ago, so the references may be older, but thanks for the comment.

Reviewer 3 Report

Thank you very much for an opportunity to review the manuscript entitled “Factor validity and generic reliability of the Developmental Coordination Disorder Questionnaire in the Czech population. The manuscript aimed to verify the factor validity of the DCDQ in a population of Czech parents whose children aged 6-10 years. Generally, the manuscript needs more revision before getting published.

Abstract

            Please check on error typing on line 25.

Introduction

1.   The introduction part is well written with a clear literature and rationale of studying in family whose children aged 6 to 10 years.

2.   Please make correction for “Taiwan” instead of “Thaiwan” line 64, reference number 17.

3.   Please clearly state the aim of the study in the introduction part.

Methodology:

1.     How the number of 850 parents with children were recruited. Was sample size calculation performed to obtain an optimal number of participants?

2.     What is the inclusion and exclusion criteria of children aged 6-10 years and those criteria of their parents? Please give more details of the criteria, especially parents (fathers or mothers, or both) were needed to fill in the questionnaire.

3.     Please write more details regarding the data cleaning which resulted in 651 participants. Why the participants were excluded? Authors wrote about the data cleaning twice on line 139 and line 146.

Results

1.     Authors please add more data on the demographic information or characteristic of participants, both of parents and children.

2.     Line 229 – 234 should be written in the discussion part, because it is the suggestion for further research study and clinical application.

3.     Please check on error typing line 336, shoed

Discussion

1.          In the limitation part, authors wrote “the results were got from well representative sample” How would authors define that the participants of this study were well representative sample? Please explain more in details.

2.          This study did not provide any information in the methods or results regarding who fill in the questionnaire, father or mother. Please clarify more why it would be interesting to have the results of the DCDQ responses of mothers and fathers independently to be able to compare how differently they perceive their children's motor skills.

In addition, please make correction since this manuscript is submitted for the consideration from “Children”, while the footnote is written as follows“Int. J. Environ. Res. Public Health 2023, 20, x FOR PEER REVIEW”

Author Response

Abstract

            Please check on error typing on line 25. Done

Introduction

  1. The introduction part is well written with a clear literature and rationale of studying in family whose children aged 6 to 10 years. 
  2. Please make correction for “Taiwan” instead of “Thaiwan” line 64, reference number 17. Corrected
  3. Please clearly state the aim of the study in the introduction part. Corrected

Methodology:

  1. How the number of 850 parents with children were recruited. Was sample size calculation performed to obtain an optimal number of participants?According to the power analysis, a Monte Carlo simulation of the data was performed, which showed that in the case of a respected test power of 0.9 and a number of three indicators, it was necessary to have at least 430 respondents (Muthén and Muthén 2010). The condition for completing the questionnaire was the age of the child 6-10 years and these children had to attend a regular primary school. - added
  2. What is the inclusion and exclusion criteria of children aged 6-10 years and those criteria of their parents? Please give more details of the criteria, especially parents (fathers or mothers, or both) were needed to fill in the questionnaire. The questionnaire could be completed by the child's father or mother. 
  3. Please write more details regarding the data cleaning which resulted in 651 participants. Why the participants were excluded? Authors wrote about the data cleaning twice on line 139 and line 146.Questionnaires that were not completely filled in or were missing certain information were discarded.

Results

  1. Authors please add more data on the demographic information or characteristic of participants, both of parents and children. Added
  2. Line 229 – 234 should be written in the discussion part, because it is the suggestion for further research study and clinical application. Moved
  3. Please check on error typing line 336, shoed. Corrected

Discussion

  1. In the limitation part, authors wrote “the results were got from well representative sample” How would authors define that the participants of this study were well representative sample? Please explain more in details. Added
  2. This study did not provide any information in the methods or results regarding who fill in the questionnaire, father or mother. Please clarify more why it would be interesting to have the results of the DCDQ responses of mothers and fathers independently to be able to compare how differently they perceive their children's motor skills. The comparison of results would be interesting, but it was not the focus of this study.
  3. In addition, please make correction since this manuscript is submitted for the consideration from “Children”, while the footnote is written as follows“Int. J. Environ. Res. Public Health 2023, 20, x FOR PEER REVIEW” Corrected.

Round 2

Reviewer 3 Report

Thank you for making a prompt revision. However the authors still did not clarify or make revision in some points  such as

I Results

  1. Authors please add more data on the demographic information or characteristic of participants, both of parents and children. Added

I cannot find any information regarding the demographic data of children and their parents, how many fathers or mothers filled in the questionnaire.

II Discussion

  1. In the limitation part, authors wrote “the results were got from well representative sample” How would authors define that the participants of this study were well representative sample? Please explain more in details. Added

I cannot find any more explanation on this part of limitation. Authors have only  moved the contents to different lines

  1. This study did not provide any information in the methods or results regarding who fill in the questionnaire, father or mother. Please clarify more why it would be interesting to have the results of the DCDQ responses of mothers and fathers independently to be able to compare how differently they perceive their children's motor skills. The comparison of results would be interesting, but it was not the focus of this study.

Authors wrote that "questionnaire could be filled in by fathers or mothers", however there is no data supports how many fathers or mothers have filled the questionnaire.

The suggestion by having the further study on "the results of the DCDQ responses of mothers and fathers independently to be able to compare how differently they perceive their children's motor skills." has been raised by the Authors. please clarify more why this point is interesting and how the point come up, which parts of result lead to this interest,  because the manuscript has no data on how many respondents were fathers or mothers.

Author Response

Response:

The reviewer is correct, the data was not taken into account in the study because it was not complete.

To the section PARTICIPANTS we added:

"The questionnaire was completed by 651 parents, of whom 485 were mothers, 104 were fathers, 23 were teachers, and 39 did not indicate their gender. In future studies, it would be beneficial to ensure complete data collection from all groups to allow for a more comprehensive comparison between mothers, fathers, and teachers."

To the section DISCUSSION we added a limitation:

While this study provides valuable insights, it is limited by the lack of information on the respondents who completed the questionnaire. In particular, it would be beneficial to distinguish between responses from mothers, fathers, and teachers to enable more detailed comparisons where relevant.
